# 2D Collagen Membranes from Marine Demosponge *Chondrosia reniformis* (Nardo, 1847) for Skin-Regenerative Medicine Applications: An In Vitro Evaluation

**DOI:** 10.3390/md21080428

**Published:** 2023-07-28

**Authors:** Eleonora Tassara, Caterina Oliveri, Luigi Vezzulli, Carlo Cerrano, Lian Xiao, Marco Giovine, Marina Pozzolini

**Affiliations:** 1Department of Earth, Environment and Life Sciences (DISTAV), University of Genova, Via Pastore 3, 16132 Genova, Italy; eleonora.tassara@edu.unige.it (E.T.); caterina.oliveri@unige.it (C.O.); luigi.vezzulli@unige.it (L.V.); 2Department of Life and Environmental Sciences, Polytechnic University of Marche, 60131 Ancona, Italy; c.cerrano@univpm.it; 3Faculty of Naval Medicine, Naval Medical University, Shanghai 200433, China; hormat830713@hotmail.com

**Keywords:** porifera, demosponges, collagen, biomaterial, skin-regenerative medicine

## Abstract

Research in tissue engineering and regenerative medicine has an ever-increasing need for innovative biomaterials suitable for the production of wound-dressing devices and artificial skin-like substitutes. Marine collagen is one of the most promising biomaterials for the production of such devices. In this study, for the first time, 2D collagen membranes (2D-CMs) created from the extracellular matrix extract of the marine demosponge *Chondrosia reniformis* have been evaluated in vitro as possible tools for wound healing. Fibrillar collagen was extracted from a pool of fresh animals and used for the creation of 2D-CMs, in which permeability to water, proteins, and bacteria, and cellular response in the L929 fibroblast cell line were evaluated. The biodegradability of the 2D-CMs was also assessed by following their degradation in PBS and collagenase solutions for up to 21 days. Results showed that *C. reniformis*-derived membranes avoided liquid and protein loss in the regeneration region and also functioned as a strong barrier against bacteria infiltration into a wound. Gene expression analyses on fibroblasts stated that their interaction with 2D-CMs is able to improve fibronectin production without interfering with the regular extracellular matrix remodeling processes. These findings, combined with the high extraction yield of fibrillar collagen obtained from *C. reniformis* with a solvent-free approach, underline how important further studies on the aquaculture of this sponge could be for the sustainable production and biotechnological exploitation of this potentially promising and peculiar biopolymer of marine origin.

## 1. Introduction

Acute and chronic skin injuries from any causes are extremely common events in both human and veterinary medicine, representing one of the highest costs issues bearing on the health care system [1]. Although they have been used for a long time as a treatment for wounds, skin grafts might not be the best solution, as their sources are limited, and they can cause intense rejection responses in patients [2]. Hence, the regenerative medicine research area is constantly seeking to develop increasingly efficient wound-dressing devices and bioengineered artificial skin-like substitutes (SLSs) [3] able to provide as much as possible collateral-effect-free treatment for the regeneration of damaged tissues, while improving and fastening the healing process. A good skin substitute should resemble the skin itself in its structure and function, especially providing a barrier against liquid loss and microbial infections, while presenting low antigenicity and maintaining good thermal-mechanical properties [4]; this has brought an ever-increasing interest in innovative polymers suitable to produce such tools. Among the biomaterials that can fit this role, collagen is one of the most widespread, and can be used (a) pristine, (b) blended with natural and/or synthetic polymers, or (c) as a coating for fibers previously obtained from other polymers [5]. To date, the most popular and economically convenient sources are rat, porcine, bovine, and chicken-derived collagens [6]; however, although it is accepted as a safe biomaterial, collagen is an animal-derived product and, consequently, has always brought up concerns about its immunogenicity [5] and ethics. In particular, mammalian collagen has been identified as a possible vehicle of transmission for bovine spongiform encephalopathy (BSE), transmissible spongiform encephalopathy (TSE), and foot-and-mouth disease (FMD), in addition to being a potential cause of allergic reactions [7]. Moreover, it can present a challenging and expensive purification process and religious limitations in Muslim, Jewish, and Hindu cultures. On the other hand, the use of recombinant human collagen—which could potentially overcome most of these problems—is restrained due to high costs and low yields of extraction [8]. In this scenario, a valid response to the ever-growing demand for collagen could come from marine sources, as they have no limitations in use for any religion and there are no reports of possible transmissible diseases from them [9]. Marine collagen is, in fact, a promising tool for wide-ranging biomedical applications, including scaffolds in tissue engineering and drug delivery systems [8,10]. Currently, marine collagen is mainly extracted from fish-processing industry by-products and bycatch [9]; however, many more remarkable sources can be found among marine invertebrates such as jellyfish, mollusks, soft corals, echinoderms, and sponges. In particular, collagen from sponges is held in high consideration because of its unique structural features [11]. Marine sponge-derived collagen has already been employed in many studies, conducted both in vitro and in vivo, especially in testing its potential as a 3D scaffold for bone regeneration experiments and bone graft applications [12,13,14,15,16]; nevertheless, there still is a gap of knowledge about these biopolymers, especially regarding their molecular characterization and the mechanisms underlying their biosynthesis [17].

Sponges (phylum Porifera) are the most ancient metazoan group still existing [18], with more than 9500 valid species described to date [19]. Despite the simplicity of their body organization, sponges can occur in a huge variety of morphologies supported by different types of structural solutions, that in a very basic way can be traced back to mineral skeletons (made of silica or calcium carbonate) and scleroproteins. The percentage of these two kinds of components can remarkably change among species, in fact, the mineral part is scarce or even absent in some of them. There are, indeed, some sponges that rely their whole-body support upon fibrous material, such as spongin or collagen. An example of this is the widespread marine demosponge *Chondrosia reniformis*: its body completely lacks endogenous mineral support and is mostly made of collagen. The mesophyll of this animal can display a very plastic behavior, appearing resting, stiffened, or creeping if subjected to different physiological conditions [20]. Additionally, the resilience and fast regeneration capacity of *C. reniformis* make this species eligible for manipulation [21]. Moreover, the tissue regeneration and the fibrogenesis process of *C. reniformis* have been studied extensively and appeared finely regulated by a set of TGF-like growth factors and TNF-like proinflammatory cytokines [22,23]. From the body of *C. reniformis*, a collagenous fibrillar suspension can be easily extracted, as described in [24], and has already been extensively studied because of its peculiarity [25,26]. Despite coming from a poikilothermic animal, these collagen extracts appear to be more thermally resistant than standard fish collagen [21,27] and shows molecular analogies with calf skin type-I collagen [28]. It is yet to be proven that the *C. reniformis* ECM fibrillar extract can be used to produce collagen-derived bioactive peptides, collagen-based hydrogels, and collagen-based cosmetic products [29,30,31]. In addition, it is possible to realize 2D-membranes whose molecular features from the same extract and its biocompatibility has already been investigated by [24], even if no further experiments have been conducted to state its suitability as an SLS for wound healing.

Therefore, the objective of this study is to enhance the understanding of the potential uses of the above-mentioned 2D membranes in treating skin injuries. By assessing their effectiveness in skin regeneration, the ultimate aim is to develop collagen films capable of treating burns and chronic wounds. To achieve this goal, we first conducted evaluations on the biodegradability, water permeability, protein permeability, and bacterial retention capacity of the 2D membranes, as these assessments were crucial to determining their efficacy in creating a controlled wound environment. Additionally, we investigated the cellular response of fibroblast cells by examining gene expression related to extracellular matrix (ECM) proteins, such as collagen and fibronectin, as well as mediators responsible for maintaining ECM equilibrium, including interleukin-1β, tumor necrosis factor, fibroblast growth factor, and metalloprotease-3.

## 2. Results

### 2.1. Chondrosia reniformis Fibrillar Extract Extraction and Purification

Using the extraction protocol described in the Materials and Methods it was possible to achieve a fibrillar suspension with a main yield of 35% ± 1.5% (SD) of the total dry weight of the sponge tissues. Therefore, from 1 g of dried *C. reniformis* is possible to obtain on average 0.35 g of ECM fibrillar extract. Considering that this suspension contains 56.81% of collagen [24], the total yield in collagen from the starting material results equal 19.9%.

### 2.2. 2D Collagen Membrane Permeability Tests

As explained in the Materials and Methods, *C. reniformis*-derived 2D-CMs permeability to deionized water and proteins was evaluated over one week (see Figure 4).

#### 2.2.1. Water Permeability Test

In the so-called “dry-wet” condition (with just one side of the membrane put in contact with water, resembling a dry wound), visual monitoring denoted that the amount of water able to pass through the 2D-CMs after the stated period was equal to zero. In the “wet-wet” conditions (with both sides of the membrane put in contact with water, recreating a moist wound), after 1 day less than 0.1% ± 0.0043% (SD) of water was able to pass through the membrane, with a main permeability of 0.002 mL/cm^2^, while after 7 days only 0.5% ± 0.0078% of water inside the upper well flowed to the lower well, with an average permeability of 0.01 mL/cm^2^ (Table 1). Therefore, is possible to consider these 2D-CMs as an effective liquid-retaining barrier.

#### 2.2.2. Protein Permeability Test

For protein permeability, less than the 0.03% (mean value: 0.01 mg/mL ± 0.0035 (SD)) of bovine serum albumin (BSA) (chosen as standard by following the example of [32]) passed through the membrane after 7 days (see Figure 1). Thus, in this range of time, the 2D-CMs are almost impermeable to proteins.

### 2.3. Bacteria Infiltration Tests

The efficiency of *C. reniformis*-derived 2D-CMs in preventing bacterial infiltration into a wound was evaluated as described in the Materials and Methods Section, by using three different bacterial species known to be common opportunistic pathogens in nosocomial environments which often cause infections in hospitalized patients [33]. The fibrillar network obtained from a 3.4 mg/mL concentrated collagen suspension was dense enough to retain almost all bacteria, reaching 100% retained bacteria for *S. aureus* in each experiment. Overall, less than an average of 0.00045% ± 0.0003 (SD) of the initiating bacteria (10^7^ cells/mL) was able to pass through the membrane (see Table 2). In all cases, bacteria in the upper layer of the membrane remained completely viable, stating that the low number, or even the absence, of CFU from the solution taken from the lower well is due to an actual scarcity/absence of bacterial cells, rather than to a mortality phenomenon of the aforementioned 2D-CMs; therefore, they acted as an extremely efficient barrier against bacteria infiltration.

### 2.4. Bacteriostaticity Test

It has already been confirmed that the membranes obtainable from the collagen extracts of *Chondrosia reniformis* have interesting properties, such as antioxidant activity [24]. Additionally, it has been observed that the liquid fibrillar ECM extract can be conserved at 4 °C for a long time without showing any sign of bacterial contamination or degradation, even if kept in non-sterile conditions (operator observations). To evaluate their potential as SLSs and consider the susceptibility of injured skin to bacterial infections, the hypothesis by which this biomaterial could be intrinsically bacteriostatic was examined. However, 2D-CMs do not seem to have any kind of effect on slowing bacterial growth, at least in the chosen experimental conditions (see the Materials and Methods). Table 3 reports the average number of bacterial CFUs left following 6 h in agitation with and without the presence of a 2D-CM. Since no significant difference was found between samples, it is possible to state that the *C. reniformis*-derived fibrillar ECM extract in the form of a 2D cm does not affect the growth rate of common bacteria such as *S. aureus, P. aeruginosa,* and *E. coli*.

### 2.5. In Vitro Biodegradability Test

An initial evaluation of the degradation rate of 2D-CMs in PBS or collagenase solution was conducted by quantifying the percentage of their weight loss over time, as described in the Materials and Methods. Figure 2A shows the percentage of weight loss of 2D-CMs over 21 days in PBS and in collagenase solution, respectively. Both types of samples exhibited the same trend of degradation, with a major weight loss within the first 7 days, which tended to slow down over time, while still slowly progressing. However, given the collagenous nature of the biomaterial, the 2D-CMs in collagenase solution reach higher levels of degradation more rapidly, even if *C. reniformis*-derived collagen tends to be refractory to the enzymatic attack of common collagenases beyond a certain level [25]. After 21 days, the samples in PBS solution on average lost 31.6% ± 0.11% (SD) of the weight, while reaching 44.2% ± 0.10% (SD) in collagenase solution in the same amount of time.

Consequently, the incubation media of the in vitro biodegradability test were analyzed as described in the Materials and Methods to better understand which kind of macromolecules were primarily released causing weight loss of the 2D-CMs. Figure 2B shows the release of total proteins within the media, while Figure 2C,D represent collagen-related weight loss and the glycosaminoglycans-related weight loss, respectively. Again, in all cases, the trend is similar for both types of samples, with higher values for those that were soaked into a collagenase solution. After 21 days, the major weight loss (up to 28.5% ± 5% (SD)) of the total weight loss in PBS samples, and up to 42.2% ± 8.5% (SD) in collagenase samples is attributable to total proteins (Figure 2B), of which collagen represents 4% ± 0.81% (SD) and 6% ± 0.9% (SD) in PBS and collagenase samples, respectively (Figure 2C). Glycosaminoglycans (GAGs), in Figure 2D, represent 13.2% ± 3.86% (SD) and 21.42% ± 2.93% (SD) of the total material released from the 2D-CMs soaked in PBS and in collagenase solution, respectively. It appears that a certain percentage of ECM components (roughly, 40% for membranes soaked in PBS solution and 20% for those in collagenase solution) were responsible for the observed weight loss of the 2D-CMs, but this not wholly identified.

### 2.6. Effect of the 2D Collagen Membranes on Fibroblasts

Fibroblasts constitute many tissues in the body and are mainly responsible for the production and turnover of ECM fundamental constituents. During tissue repair processes, fibroblasts can assume a contractile phenotype involved both in increased ECM production and contraction phenomena [34]. In a normal wound healing process, ECM molecules need to be rapidly synthesized for effective tissue remodeling, and their biosynthesis must be finely regulated. Excessive deposition of connective tissue could generate hypertrophic scars and keloids; therefore, a correct balance between connective tissue synthesis and breakdown is strictly necessary. Normally, this balance is controlled by the production of mediators like cytokines and growth factors, such as transforming growth factor (TGF-β) family, interleukins (such as IL-1β), tumor necrosis factor (TNF), the fibroblast growth factor (FGF) and others [35]. To evaluate the response of fibroblasts in the presence of a 2D cm scaffold and, in particular, to assess if the 2D-CMs could induce the up-regulation of ECM production-/remodeling-related genes, α1-chain of collagen type I (COL1A1), fibronectin and metallopeptidase-3 (MMP3) gene expression profiles were evaluated by qPCR in L929 mouse fibroblasts growing on plates coated with *C. reniformis*-derived ECM fibrillar extract. On the same cells, a gene expression analysis was also performed to evaluate any kind of imbalance in the above-mentioned ECM regulating factors comparing them to the controls. Table 4 shows the gene expression levels of fundamental ECM-related genes in L929 fibroblast growing on 2D-CMs for 24 or 120 h. The gene expression levels of controls were considered normalized to 1. Gene expression of COL1A1, FGF, MMP3, and IL-1 does not show significant differences between controls and 2D cm coated samples, neither after 24 h nor after 120 h. Conversely, fibronectin gene expression after 120 h, equal to 1.46 ± 0.13 folds, increases significantly, showing a 46% enhancement when compared to the uncoated controls. Instead, TGF-β gene expression after 24 h, equal to 0.79 ± 0.05 folds, is lower in 2D cm coated samples than in uncoated controls, and diminishes by 21%. 

## 3. Discussion

Collagen is one of the most employed natural polymers for the production of skin substitutes. It has been reported that some collagens of marine origin display unique chemical and physical properties [36] and are suitable for regenerative medicine [37]. Among them, collagen from marine sponges has been previously suggested as a viable biomaterial for constructing scaffolds intended for bone regeneration [14]. Conversely, there is still a lack of knowledge about the usage of Porifera-derived collagens to produce skin substitutes.

Our study aimed to evaluate the suitability of 2D membranes obtained using an intact fibrillar extract derived from the marine sponge *C. reniformis* for skin regenerative medicine purposes. Using the extraction procedure described in the Materials and Methods Section, it was possible to averagely obtain 0.35 g of ECM fibrillar extract from 1 g of dried sponge tissue. This result indicates that increasing the number of extraction cycles from two, as previously described [24], to six, i.e., until reaching the complete dissolution of the sponge body, does not seem to increase the extraction yield. The collagen fibers obtained in the first isolation cycles may be more insoluble and easier to recover than the ones extracted in the last cycles, and, therefore, could have been lost during the repeated washing steps. As stated in Section 2.1, the total yield of collagen starting from fresh tissue is equal to 19.9%. The isolation of collagen from the same organism using a carbon dioxide extraction approach provided a yield of only 10%, aside from it being faster [38]. Given the richness of the collagen in this animal, with the solvent-free extraction procedure described here it is possible to obtain an overall good quantity of intact collagen fibers. Being a purification method based on trypsin digestion and scaling up the extraction system with a view to a develop a circular economy, it could be possible to replace the more expensive purified trypsin with fish stomach extracts deriving from waste material from fish processing.

When 2D membranes are designed as skin substitutes, they must ensure limited evaporation of water and loss of proteins from the injured area. For the *C. reniformis* derived 2D-CMs, no water passage was registered in the “dry-wet” condition, whereas when a moist wound was recreated in the “wet-wet” conditions a main permeability of only 0.01 mL/cm^2^ was detected after 7 days. The water permeability across the collagen membranes is affected by porosity, pores diameter, cross-links level, and cross-links hydrophilicity. An increased diffusion coefficient is related to a reduced cross-linking state [39]. The strong water insolubility observed in the *C. reniformis* fibrillar extract is evidence of a high level of interchain crosslinking, which allows the obtaining of 2D membranes through directly casting the collagen extract without any chemical crosslinking steps. The high crosslinking rate gives this biomaterial a reduced permeability to water. The water permeability rate detected in *C. reniformis* collagen-derived membranes was about three-fold lower than those evaluated in 2D membranes derived from sea urchin collagen [32]. Although these two types of membranes are similar in composition, the reduced water permeability detected in this study can be ascribed to a higher polymer concentration used for casting the membranes. Even if a biomaterial for wound dressing must guarantee a limited loss of water during healing, a certain permeability level is necessary, as excessive water retention may cause wound exudate accumulation [40]. Therefore, to obtain optimal water permeability performance in 2D membranes designed as skin substitutes, the polymer concentration and, accordingly, its porosity level must be modulated properly.

Moreover, *C. reniformis*-derived 2D-CMs showed low permeability to proteins, preventing the loss of growth factors, and ensuring maintenance of the optimal microenvironment in the regeneration region. *C. reniformis*-derived 2D-CMs proved to be a strong barrier against bacterial infiltration, against both Gram-positive and Gram-negative strains. This seems to be particularly evident for Gram-positive strains, where the percentage of bacteria retention is 100%. Although the sponge-derived 2D-CMs constitute a thick barrier that prevents the infiltration of bacteria into the injured area, the contact of the tested microorganisms with the membranes’ surface did not affect their viability; furthermore, the 2D-CMs do not seem to release any soluble substance with a bacteriostatic activity. Sponges are populated by numerous symbiotic microorganisms [41]; therefore, across evolution, they had to develop a manifold molecular strategy to contain bacteria over-growth, and, for this reason, marine sponges are important sources of natural compounds with antibiotic and antiviral action [42]. However, the extraction procedures used in the present study allowed us to obtain an extract of intact collagen fibers, free from any sponge-derived secondary metabolites. As poriferan natural compounds exhibit antitumor and cytotoxic activities [43], the previously attested biocompatibility of 2D-CMs on in vitro cell lines [24] further confirms this issue.

Biomaterials for tissue engineering have to display a low degradation rate but, on the other hand, once the regenerative process is finished, they must be resorbable. Previous studies have demonstrated the peculiar resistance of *C. reniformis*’ collagen extract to collagenase digestion [25]. Furthermore, after 6 days in collagenase solution, the *C. reniformis* derived 2D-CMs remained intact if compared to commercial membranes [24]. In this study, a 2D-CMs time-course stability test within 21 days was performed. Although within 21 days of incubation the sponge-derived 2D-CMs seemed to maintain their integrity, a certain percentage of weight loss was detected in both saline solution and collagenase solution. The higher degradation rate observed in collagenase-treated samples indicates that this biomaterial is not completely resistant to enzyme digestion and, hence, is potentially resorbable. A remarkable contribution to membrane weight loss derived from GAGs: the amount of glycosaminoglycans released in the medium was constant and directly proportional to the weight loss along the entire observation time, as well as being higher in the presence of collagenase activity. This suggests that these sulphurated polysaccharides are intimately associated with collagenous proteins that, once enzymatically attacked, cause the GAGs release. If the GAGs adhesion to the 2D-CMs is subjected to collagen proteins, the collagen release is also affected by GAGs. After seven days, when the main percentage of GAGs was released, a significant increase in collagen and non-collagen proteins was detected. It was experimentally shown that the addition of GAGs to collagen scaffolds can preserve them from collagenase digestion, improving the stability of the biomaterial [44]. *C. reniformis*-derived 2D-CMs are naturally GAGs decorated, and this explains their peculiar stability to collagenase treatment. As stated in the Results Section, a certain percentage of components in the 2D-CMs related to their weight loss during the stability tests was not characterized, but we suggest that, reasonably, it could be constituted by polysaccharidic elements; however, until a complete molecular characterization of the *C. reniformis*-derived ECM extract is reached, it will not be possible to perform specific tests to accurately clarify the nature of the above-mentioned elements.

The biocompatibility of *C. reniformis*-derived 2D-CMs has been previously attested in vitro on cell lines [29]. Additionally, in this study, an expression profile of some key genes of the wound healing process was created. During skin wound healing various matrix metallopeptidase proteins (MMPs) are involved in ECM degradation [45]. Matrix metallopeptidase 3 (MMP-3) is considered essential for the initial phase of tissue repair [46]. Here, in the L929 fibroblasts cell line, the MMP-3 gene expression was not significantly affected, indicating that the sponge-derived biomaterial does not interfere with the regular process of skin regeneration. Typically, during the recovery processes of skin injuries, ECM components such as collagens and fibronectin are released [47]. An imbalance in collagen deposition during wound healing may induce scar tissue formation over time [48]. When fibroblast cell lines were left to grow on *C. reniformis*-derived 2D-membranes, no significant difference in collagen (COL1A1) gene expression was observed for both the incubation times of 24 and 120 h, confirming how the contact of mouse fibroblasts with sponge collagen extract does not affect the fibrogenesis process preventing the risk of scaring process. Comparing these results with gene expression profiles of L929 cell lines growing on other sponge-derived 2D-CMs, the same response was observed when the mouse fibroblasts were let grow on *S. foetidus*-derived 2D-CMs, while a different response was detected in fibroblast grown on *I. oros* derived-2D-CMs, as here the contact with the biomaterial induced after 24 h a significant collagen up-regulation. These data suggest that fibrogenesis mediated by contact with these types of materials is extremely heterogeneous and linked to the species-specific molecular composition and structure of the tested biomaterial. Conversely, after 120 h, L929 mouse fibroblasts growing on sponge-derived 2D-CMs showed a slight but significant up-regulation of the fibronectin gene. One of the main factors able to induce fibronectin up-regulation is tumor growth factor (TGF-β) [49]; however, this seems not to be the case, as TGF-β expression was low during the first stage of the experiments. Instead, fibronectin up-regulation could be here related to the interaction of cells with marine collagen. Collagens, including sponges’ collagens, are known for containing RGD fibronectin-binding sites [50] and the presence of this motif is confirmed in the fibrillar collagen of *Chondrosia reniformis* in which the aminoacidic sequence has already been partially described [51]. The RGD motif is an important recognition site found in several fibrillar collagens, including collagen types I, II, and III. This motif is recognized by integrin receptors, particularly integrin α5β1. The binding of integrin α5β1 to the RGD motif can activate intracellular signaling pathways to induce epithelial-mesenchymal transition phenomena typical of the wound healing process [52,53]. As collagen fibronectin is a key ECM protein released during the early phases of skin repair [54], the interaction with 2D-CMs could be considered able to stimulate the wound-healing process in fibroblast cell lines.

Finally, 2D-CMs seem not to affect the expression of neither FGF or cytokines 4genes such as IL-1β that are involved in skin wound healing response [55]. These results indicate that, although *C. reniformis* derived 2D-CMs when compared to commercial collagen membranes were able to improve L929 cells proliferation within the first 3 days of incubation [24] and fibronectin gene expression after 120 h, these responses are not under the control of the above-mentioned growth factors and cytokines.

## 4. Materials and Methods

### 4.1. Chemicals

All reagents were acquired from SIGMA-ALDRICH (Milan, Italy) unless otherwise stated.

### 4.2. Sponge Sampling

Several specimens of *Chondrosia reniformis* were harvested in the waters of the Portofino Promontory (Liguria, Italy) at 10–20 m of depth by scuba diving, then maintained in thermic bags at a temperature of 14–15 °C until arrival in the laboratory, where the animals were allowed to acclimatize in an aquarium. Stabulation was conducted by keeping the animals at 14 °C in a 200 L aquarium, containing natural seawater collected in the same area of the sampling and equipped with an aeration system, with a salinity of 38‰.

### 4.3. Chondrosia reniformis Fibrillar Collagen Extraction and Purification

A pool of fresh animals, weighing a total of 70 g, was processed to obtain the collagen fibrils in their native conformation as described in [56], with some modifications. In short, sponges were rinsed twice with artificial seawater, cut into small pieces of about 4–5 mm^3^, and put into 5 volumes of a solution with 0.1% trypsin (Sigma-Aldrich, Milan, Italy) and 100 mM ammonium-bicarbonate buffer (pH 8.5), then left overnight at 37 °C into a horizontal shaker. The dark liquid deriving from enzymatic digestion was removed by filtration with a metal strainer, while the solid material was suspended in 3 volumes of cold deionized water and incubated at 4 °C for three days in a rotating-disc agitator. The viscous suspension thus obtained was filtered with a metal strainer (keeping the remaining fragments of tissue for further cycles of extraction) and firstly centrifuged at 1500× *g* for 10 min at 4 °C, to remove any residues of cells and sediment particles. The supernatant fluid containing the collagen suspension was finally recovered by centrifugation at 18,000× *g* for 35 min at 4 °C. The resulting pellet was washed twice with deionized water to eliminate any trypsin residues and resuspended in clean deionized water. This entire procedure was repeated six times until the fragments of sponge tissue were completely dissolved. The final fibrillar suspension, resulting from all the collagen collected from extraction cycles, has been stored at 4 °C until use. To improve the biocompatibility of the sample, some of the polysaccharidic components—which are co-extracted with sponge collagen [24]—were partially removed as explained in [57], by simply treating it with an equal volume of 0.1 M NaOH for 6 h at 25 °C, then recovering the collagen suspension by centrifugation at 18,000× *g* for 35 min, with two further washing steps. The ECM extract concentration (mg/mL) was then determined by drying and weighing a known volume of the suspension. The total collagen yield was calculated considering the previous observations provided by [24].

### 4.4. Production of 2D Collagen Membranes

The 2D collagen membranes (2D-CM) were obtained by pouring 3 mL of a 3 mg/mL ECM fibrillar extract (ECM-FE), dissolved in distilled water, in rectangular silicon molds (25 × 28 mm) and left overnight at 37 °C to dry completely, until thin bidimensional sheets with a mean weight of 8.5 ± 2.7 mg and a thickness of 0.1 mm were obtained, which were used for permeability, bacteria infiltration, bacteriostatic activity, and in vitro biodegradability tests. For gene expression analysis, the ECM-FE was used to directly coat 6-well plates. Each 2D cm coating was sterilized under UV light for 30 min.

### 4.5. 2D Collagen Membranes Permeability Tests

The ability of the *C. reniformis*-derived 2D-CMs to avoid liquid and protein loss was evaluated by testing their permeability to both distilled water and bovine serum albumin (BSA). The 2D-CMs were fixed into specific, ad hoc created supports (used as shown in Figure 3), every part of which was sterilized in an autoclave before assembling and use; the effectiveness of the experimental setup was confirmed with some preliminary trials. The experiments, performed in triplicate at room temperature, followed the example of [32] and were meant to recreate a dry and a moist skin wound condition. In the former case, the term “dry” indicates a type of skin injury that does not release exudates; therefore, it was reproduced a situation in which one side of the 2D cm was exposed to air and the other side was wetted (“dry-wet” condition). In the latter case, the word “moist” denotes a type of skin injury that releases exudates; here, both sides of the 2D cm were hydrated (“wet-wet” condition). To prevent long-term evaporation, the structures were, in all cases, kept in a humidified environment.

#### 4.5.1. Water Permeability Test

“Dry-wet” condition: 2 mL of deionized water was put into the upper well, while the underlying well was left empty (see Figure 4). The passage of water through the 2D cm was visually monitored at 0, 1, 3, 6, 24, 48 h, and 7 days, respectively.

“Wet-wet” condition: 2 mL of deionized water was put into the upper and the lower wells, so that both sides of the 2D cm were in contact with water, as in Figure 4. The water passage through the 2D-CMs was quantified by weighing the water in the upper well at the same time points selected for the “dry-wet” condition. To prevent evaluation biases, the inserts were kept in a closed environment, and a control insert was used, which allowed us to keep track of possible evaporation. The percentage of deionized water that crossed the 2D-CMs after the stated time points was obtained as follows: [weight (g) of water in the upper well at a final time point/weight (g) of distilled water in the insert at time zero] × 100. The permeability of the 2D-CMs was expressed as the volume of water passing through the surface at each time-point (mL/cm^2^).

#### 4.5.2. Protein Permeability Test

In the “dry-wet” experiment condition with distilled water, no amount of liquid could pass through the membrane; this stated that a solute cannot cross a barrier if its solvent is not able to do so in the first place. Only the “wet-wet” condition experiment was performed for the protein permeability test, as in Figure 4. The upper well was filled with 2 mL of a 35 mg/mL bovine serum albumin (BSA) solution prepared in 50 mM PBS (pH 7.4), while the lower well contained just deionized water. At specific time points (1, 3, 6, 24, 48 h, and 7 days, respectively), 100 μL of the lower well content was taken and used to perform a Bradford assay [58] to quantify the protein amount in the solution. Samples were read at a 595 nm wavelength using a Beckman spectrophotometer (DU 640, (Beckman Coulter s.r.l., Milan, Italy). For each time-point, the amount of BSA occurring in the lower well (representing the protein permeability of the 2D-CMs) was calculated as a percentage as follows: 35 mg/mL: 100% = [BSA] in the lower wells: x.

### 4.6. Bacteria Infiltration Tests

Aside from the capacity to prevent water and protein loss, the epidermal layer should be able to keep pathogens from entering the body. The 2D-CMs were therefore tested to evaluate their effectiveness in avoiding bacteria infiltration. Three different strains of bacteria were used: *Staphylococcus aureus* ATCC 29273 (Gram-positive coccus), *Pseudomonas aeruginosa* ATCC 27853, and *Escherichia coli* ATCC 25404 (both Gram-negative bacilli). The 2D-CMs were fixed to the same type of support as created for water and protein permeability tests, as explained in Section 4.4. and shown in Figure 3. The upper well was filled with 2.5 mL of a 1× PBS solution (not supporting bacteria growing conditions) containing bacteria at a concentration of 10^7^ cells/mL, while the lower well only contained the 1× PBS sterile solution. The two solutions could get in touch only through the membrane. The structures were left in incubation for 48 h at 37 °C. After 48 h, both the upper and lower wells content was collected to be analyzed (the former, to verify bacteria viability; the latter, to check an eventual bacterial infiltration through the 2D-CM). The solutions were centrifuged for 10 min at 5000× *g* to pellet any bacteria; then, the pellet was re-suspended in 2.5 mL of 1× PBS. Finally, 100 µL of the samples were plated on LB agar plates, some undiluted and several-diluted in 1× PBS (10^−3^ to 10^−6^ dilutions), and left in incubation overnight at 37 °C. The number of colony-forming units (CFUs) was determined for each sample. The permeability of the 2D-CMs to bacteria was expressed as the percentage of bacteria found in the flow-through compared to the initial bacteria concentration (10^7^ cells/mL). Accordingly, the percentage of retained bacteria was also calculated. The same procedure was carried out in triplicate for all the selected bacterial species.

### 4.7. Bacteriostaticity Test

The effect of *C. reniformis*-derived 2D-CMs was tested on the growth of *Staphylococcus aureus* ATCC 29273, *Pseudomonas aeruginosa* ATCC 27853, and *Escherichia coli* ATCC 25404 to evaluate if they could show bacteriostatic activity. One 25 × 28 mm membrane was put inside a tube filled with 10 mL of an LB solution with a bacterial concentration of 1 × 10^4^ CFUs/mL and left in moderate agitation at 37 °C for 6 h, which was considered enough time for the bacteria to reach their exponential growth phase, as well as for 24 h. Control tubes with any membranes were also prepared. After the expected amount of time, 100 μL of each sample was plated, undiluted or several-diluted (10^−1^ to 10^−4^ dilutions), into LB agar plates and left in incubation overnight to monitor any difference in the CFU number between samples. The same experiment was repeated in triplicate. 

### 4.8. In Vitro Biodegradability Test

A good biomaterial for SLSs should be thermally and mechanically stable, but it should also be biodegradable in the medium/long term. Thus, the stability of the 2D-CMs was evaluated in vitro. The weight (mg) of the dry membranes was registered (W_i_), then they were soaked inside wells containing 5 mL of 1× PBS (pH = 7.4) solution or of a 0.1 mg/mL *Clostridium histolyticum* collagenase (Sigma-Aldrich, Milan, Italy) solution prepared with 50 mM tricine, 10 mM CaCl_2_, and 400 mM NaCl and left in incubation at 37 °C for 7, 14, and 21 days, respectively. Collagenase was renewed every 7 days. After the expected amount of time, the 2D-CMs were air-dried, and their weight was annotated again (W_f_). The percentage (%) of average weight loss was expressed as follows: [(W_f_ − W_i_)/W_i_] × 100.

#### Evaluation of the Material Released into the Incubation Media

The incubation media, in which the 2D-CMs were soaked for the in vitro biodegradability test, was recovered at 7, 14, and 21 days and used to evaluate the organic material released from the membranes, responsible for their weight loss over time.

The protein component released in the media from the 2D-CMs in PBS or collagenase solution was measured using the bicinchoninic acid method [59] after 7, 14, and 21 days using PBS or collagenase solution as a blank, respectively.

Since the ECM-FE from which the 2D-CMs were made with mainly constitutes of fibrillar collagen and ECM-associated glycosaminoglycans, a hydroxyproline assay and a GAGs assay were also performed. To quantify the percentage of collagen-deriving material released in PBS or collagenase solution from the 2D-CMs during the experiment, the incubation media were hydrolyzed in NaOH 2 M at 120 °C for 20 min at 1 atm, then the hydroxyproline concentration was evaluated as described in [60]; the absorbance of each sample was read with Beckman spectrophotometer (DU 640), wavelength 550 nm, using a cis-4-hydroxy-l-proline as a standard curve. For each sample, the degraded collagen content was obtained using the proportion factor of 1 g of hydroxyproline per 6 g of collagen, as stated in [27]. Finally, the GAGs content within the incubation media was quantified as described in [61]. The absorbance of each sample was read with a Beckman spectrophotometer (DU 640), wavelength 620 nm, using a standard curve of 0.1 mg/mL chondroitin sulfate.

### 4.9. Effect of 2D Collagen Membranes on Fibroblasts

#### 4.9.1. Cell Cultures

The L929 mouse fibroblast cell line was obtained from the National Collection of Type Cultures (NCTC, Salisbury, UK). Cell cultures were maintained at 37 °C in a humidified, 5% CO_2_ atmosphere, in high glucose Dulbecco’s modified Eagle’s medium (D-MEM) with glutamax (Euroclone, Milan, Italy), which was supplemented with 10% FBS (Euroclone) and added with penicillin/streptomycin as antibiotics.

#### 4.9.2. Fibroblast Gene Expression Analysis

To perform a further gene-expression analysis, L929 mouse fibroblast cells were seeded into 6-well plates that were or were not pre-coated with the *C. reniformis*-derived ECM-FE, as described in Section 4.3. Cells were seeded at a density of 300,000 cells/well for the 24 h gene expression analysis or 50,000 cells/well for the 120 h gene expression analysis and then incubated at 37 °C for the established amount of time. At the end of each experiment, total RNA was extracted with the RNeasy Mini Kit, (Qiagen, Milan, Italy) following the manufacturer’s instructions. The cDNA was synthesized using the Revert Aid Reverse Transcriptase (Thermo Fisher Scientific, Milan, Italy) by taking 1 µg of purified total RNA from each sample. Every PCR reaction was performed in 15 µL containing 1× master mix iQ SYBR^®^Green (Bio-Rad, Milan, Italy), 0.2 µM of each primer, and 3 µL of a 1:5 diluted reverse transcription reaction buffer. Each sample was analyzed in triplicate. The following thermal conditions were used: initial denaturation at 95 °C for 3 min, followed by 45 cycles with denaturation at 95 °C for 15 s, and annealing and elongation at 60 °C for 60 s. Fluorescence was measured at the end of each elongation step. GAPDH was used as a reference gene and the values were normalized to its mRNA expression. All the PCR primers (Appendix A) were designed through the Beacon Designer 7.0 software (Premier Biosoft International, Palo Alto, CA, USA) and acquired from TibMolBiol (Genova, Italy). Data analyses were obtained from the DNA Engine Opticon^®^3 Real-Time Detection System Software program (3.03 version) and, to calculate the relative gene expression compared to an untreated (control) calibrator sample, the comparative threshold Ct method was used within the gene expression analysis for iCycler iQ Real-Time Detection System Software^®^ (2004 Bio-Rad, Milan, Italy). Data are means ± S.D. of three independent experiments performed in triplicate.

## 5. Conclusions

For the first time, described here is the effectiveness of a collagen extract derived from a marine sponge as an eligible biomaterial for the production of 2D membranes applicable in the skin regenerative medicine field. Considering the inherent limitations of the natural origin of the starting biopolymer, such as slight interspecific variability and non-modulable fiber dimensions, 2D-CMs derived from *Chondrosia reniformis*’ collagen exhibit promising potential as SLSs. However, further analyses are required, particularly in vivo or on regenerated tissues, to enhance their functionalization and ensure their suitability for practical applications.

With the formulation analyzed in this study, the 2D-CMs display very low water and protein permeability; for this reason, by modulating the concentration of the collagen extract and therefore the frame texture of the membranes, it should be possible to achieve the right compromise between optimal moisture and adequate draining of wound exudates. Moreover, though they lack any bacteriostatic activity, they appeared to be a strong barrier against both Gram-positive and Gram-negative strains. The limited degradation rate guarantees good long-term resistance, while in vitro tests attest that contact with the 2D-CMs does not induce the formation of scar tissue or inflammatory processes over time.

These data, combined with the high extraction yield obtained with a solvent-free approach, underline how further studies on the aquaculture of this sponge are important for the sustainable production of this potentially promising and peculiar biopolymer of marine origin.

## Figures and Tables

**Figure 1 marinedrugs-21-00428-f001:**
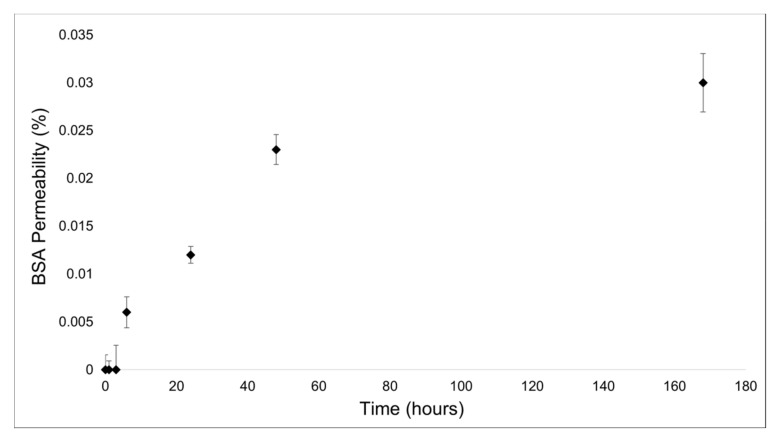
Percentage of bovine serum albumin (BSA) passing through the 2D collagen membranes at specific time points. Time is expressed in hours. The graph shows mean values ± standard deviation (SD).

**Figure 2 marinedrugs-21-00428-f002:**
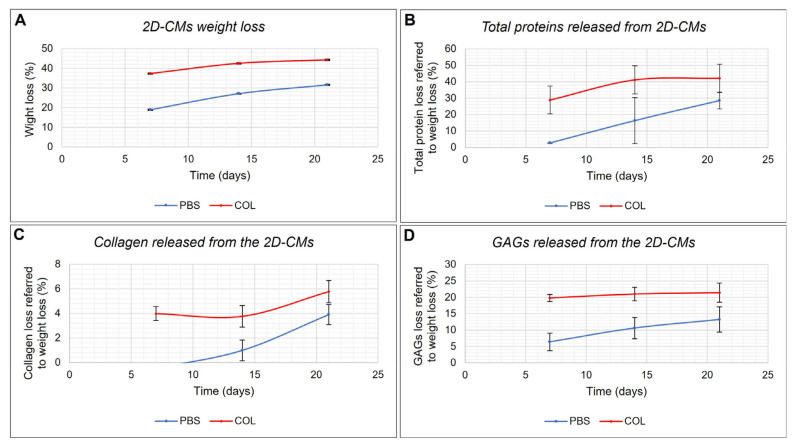
In vitro biodegradability test. (**A**) Percentage of degradation of the 2D-CMs in PBS and in 0.1 mg/mL collagenase solution calculated as 2D-CMs dry weight difference. The incubation media were used to determine the percentage of total proteins released from the 2D-CMs (**B**), the percentage of collagen release (**C**), and the percentage of GAGs release (**D**) referred to the total weight loss of the membranes.

**Figure 3 marinedrugs-21-00428-f003:**
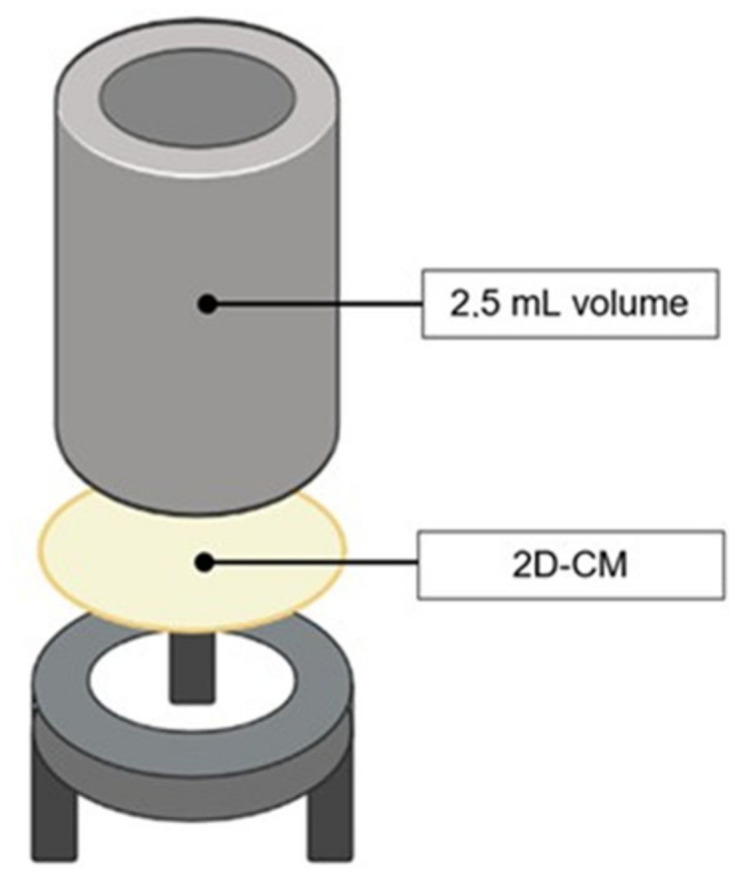
Structure of the tools used for permeability and bacteria infiltration tests.

**Figure 4 marinedrugs-21-00428-f004:**
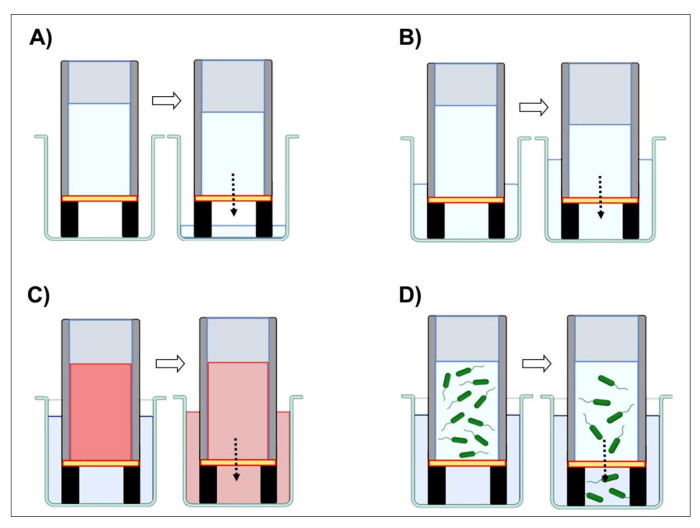
Experimental setup for the *C. reniformis*-derived 2D-CMs permeability tests. (**A**) Permeability to deionized water in “dry-wet” conditions. (**B**) Permeability to deionized water in “wet-wet” conditions. (**C**) Permeability to proteins (BSA) in “wet-wet” conditions. (**D**) Bacteria infiltration test.

**Table 1 marinedrugs-21-00428-t001:** Percentage (%) of water able to pass through the 2D-CMs, with relative standard deviations (SD), at different time points and in different experimental conditions.

Time Point	“Dry-Wet” Condition	“Wet-Wet” Condition
0 h	0	0
1 h	0	0
3 h	0	0
6 h	0	0
24 h	0	0.1 ± 0.0043
48 h	0	0.15 ± 0.27
7 days	0	0.5 ± 0.0078

**Table 2 marinedrugs-21-00428-t002:** *S. aureus*, *P. aeruginosa*, and *E. coli* infiltration. As described in the Materials and Methods, the bacterial suspensions were loaded onto the 2D-CMs at a concentration of 10^7^ CFUs. The numbers indicate the CFUs that were recovered in the flow-through, therefore representing the bacteria that successfully crossed the 2D-CMs within a 48 h period. The percentage (%) of infiltrated bacteria, along with their respective standard deviations (SD), must be considered in relation to the initial bacterial concentration of 10^7^ CFUs used in the experiments.

Experiment	*S. aureus*	*P. aeruginosa*	*E. coli*
1st	0	0	0
2nd	0	30	20
3rd	0	10	120
**Mean**	0	20	70
**Infiltrated bacteria (%)**	0	0.0002	0.0007
**±SD**	0	0.000153	0.000643
**Retained bacteria (%)**	100	99.9998	99.9993

**Table 3 marinedrugs-21-00428-t003:** Evaluation of the bacteriostatic activity of 2D-CMs on *S. aureus, P. aeruginosa,* and *E. coli* after a 6 h incubation in LB media, expressed with relative CFUs/mL registered at the end of the experiment.

Sample	*S. aureus*	*P. aeruginosa*	*E. coli*
Control	5.6 × 10^8^ ± 1.9 × 10^8^	1.28 × 10^8^ ± 1.9 × 10^8^	1.41 × 10^8^ ± 2.08 × 10^8^
Presence of 2D-CM	6.1 × 10^8^ ± 1.6 × 10^8^	1.9 × 10^8^ ± 0.7 × 10^8^	1.47 × 10^8^ ± 2.41 × 10^8^

**Table 4 marinedrugs-21-00428-t004:** The gene expression level of some of the main ECM-related genes: α1-chain of collagen type I (COL1A1), fibronectin (FN), metallopeptidase-3 (MMP3), transforming growth factor (TGF-β), fibroblast growth factor (FGF), and interleukin 1β (IL-1β) on L929 mouse fibroblasts growing on *C. reniformis*-derived 2D-CMs for 24 and 120 h. Significant differences between gene expression levels were calculated by comparing the two types of samples to an uncoated control. Asterisks indicate a significant difference versus the respective control (paired Tukey test, * = *p* < 0.05).

*Sample*	COL1A1	FN	MMP3	TGF-β	FGF	IL-1β
**24 h**	0.94 ± 0.05	1.23 ± 0.2	1.25 ± 0.23	0.79 ± 0.05 *	0.66 ± 0.28	1.08 ± 0.21
**120 h**	1.27 ± 0.20	1.46 ± 0.13 *	1.20 ± 0.55	0.94 ± 0.24	0.76 ± 0.16	1.30 ± 0.25

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
