# Peer review of "2D Collagen Membranes from Marine Demosponge Chondrosia reniformis (Nardo, 1847) for Skin-Regenerative Medicine Applications: An In Vitro Evaluation"

_marinedrugs, 2023, doi:10.3390/md21080428_

Round 1

Reviewer 1 Report

Tassara et al. fabricated a 2D-membrane with a collagen extract derived from a marine sponge called Chondrosia reniformis. The authors investigated the formation process, permeable abilities, in vitro biodegradability and antibacterial property of as-prepared membrane. After careful consideration, I reckon this article can be published in this journal after a major revision.

 Comment:

1.     There are too many grammatical mistakes. Please thoroughly and carefully check the manuscript and revise all of them.

2.     Introduction, Page 2 of 17: when mentioning “innovative polymers suitable to produce appropriate SLSs”, references describing advantages of hydrogels should be mentioned. For example, papers like: “High-strength hydrogels: Fabrication, reinforcement mechanisms, and applications”, Nano Research. 16, 3475–3515 (2023).

3.     To all text: please check all the orders of your citation numbers. Normally, the number of previously cited article should always appear before the newly cited one. In Page 2 of 17, ‘[9,7]’ and ‘[26,20]’ are wrongly cited.

4.     Introduction, Page 3 of 17: the final part of introduction is too brief for readers to initially grasp the context of the full text. Here, enlarging this paragraph by depicting some processes and results of your experiments are highly recommended here. Of course, your can also illustrate the possible prospects of your prepared 2D-menbrane in the realms of medicine and biology.

5.     Could you please explain why such a great disparity will occur within the three experiments conducted on E.coli?

6.     Chapter 4.2, Page 10 of 17: the formats of the two formulas are inappropriate, which may lead to unnecessary confusion and misunderstanding.

7.     Chapter 4.4.2, Page 11 of 17: could you explain more on the reasons why you only chose “wet-wet” condition for protein permeability test, since the connection of limited water permeabilty with protein permeability is not that obvious for me. A more powerful and convincing conclusion may be drawn through experiments under both “dry-wet” and “wet-wet” conditions.

The english language of the manuscript should  be improved.

Author Response

Point-by-point reply to Reviewer #1

Tassara et al. fabricated a 2D-membrane with a collagen extract derived from a marine sponge called Chondrosia reniformis. The authors investigated the formation process, permeable abilities, in vitro biodegradability and antibacterial property of as-prepared membrane. After careful consideration, I reckon this article can be published in this journal after a major revision.

Comment:

  1. There are too many grammatical mistakes. Please thoroughly and carefully check the manuscript and revise all of them.

R: We thank the reviewer for pointing out this shortcoming. We checked the whole manuscript and fixed mistakes and misspellings in the text.

  1. Introduction, Page 2 of 17: when mentioning “innovative polymers suitable to produce appropriate SLSs”, references describing advantages of hydrogels should be mentioned. For example, papers like: “High-strength hydrogels: Fabrication, reinforcement mechanisms, and applications”, Nano Research. 16, 3475–3515 (2023).

R: We added the reference as suggested; nevertheless, in this work, we proposed a xerogel, and not a hydrogel, as a potential skin-like substitute device.

  1. To all text: please check all the orders of your citation numbers. Normally, the number of previously cited article should always appear before the newly cited one. In Page 2 of 17, ‘[9,7]’ and ‘[26,20]’ are wrongly cited.

R:We fixed the order of citation numbers, as suggested.

  1. Introduction, Page 3 of 17: the final part of introduction is too brief for readers to initially grasp the context of the full text. Here, enlarging this paragraph by depicting some processes and results of your experiments are highly recommended here. Of course, your can also illustrate the possible prospects of your prepared 2D-membranes in the realms of medicine and biology.

R:The final part of the introduction was enriched as suggested, also including future prospects related to the potential application of the 2D-CMs.

  1. Could you please explain why such a great disparity will occur within the three experiments conducted on  coli?

R: The results obtained with E. coli in the infiltration test may appear discordant with each other, but they have to be referred to the initial number of CFUs that we put on the other side of the membranes. On the upper side of the membrane, there was a bacterial suspension of 107 cells/mL; 20 and 120 represent respectively the 0.0002% and the 0.0012% of the initial concentration, therefore (even if there’s a ten-factor difference) the difference between them was considered not significant in comparison with the starting concentration. To clarify this aspect, we also enriched the description of the table that summarizes these results (Table 2).

  1. Chapter 4.2, Page 10 of 17: the formats of the two formulas are inappropriate, which may lead to unnecessary confusion and misunderstanding.

R: To avoid misunderstandings, we decided to delete the formulas in this part of the manuscript.

  1. Chapter 4.4.2, Page 11 of 17: could you explain more on the reasons why you only chose “wet-wet” condition for protein permeability test, since the connection of limited water permeabilty with protein permeability is not that obvious for me. A more powerful and convincing conclusion may be drawn through experiments under both “dry-wet” and “wet-wet” conditions.

R: In the “dry-wet” condition, no amount of water was able to cross the membrane. As proteins were solubilized in water and stated that no amount of solute would have been able to pass through the membrane if its solvent wasn’t able to do it in the first place, we decided not to perform the “dry-wet” condition using the protein solution. We clarified this aspect by adding the information in the methods paragraph.

Reviewer 2 Report

Dear editor

 The study addressed at the potential use of a ECM fibrilar-derived 2D membrane extracted from C. reniformis (marine sponges) for wound healing applications. For this purpose, the authors assessed the biodegradability, permeability to water, proteins, and bacteria, and cellular response in fibroblast cell lines of the membranes. The data obtained in this study show na interesting extraction yield of approximately 19%, and suggest that the membrane is almost impermeable to proteins and can work as a efficient barrier against liquid loss and bacteria infiltration, although it does not affect the growth rate of bactéria commonly found in human skin surfaces. The 2D-CM is also able to modulate ECM production-/remodeling-related genes expression, such as up-regulation fibronectin and down-regulation of TGF-β genes. The study is relevant and well designed, with promising results for the further development of wound healing in vivo studies and clinical trials.

 Here are my comments and suggestions.

 1) Some relevant information needs to be added to the abstract, such as results related to potential ECM production-/remodeling-related genes modulation.

 2) In the introduction section, the authors have mentioned that there is still gap of knowledge about the molecular characterization and the mechanisms underlying the biosynthesis of the biopolymers that have been reported in wound healing studies. However, no physical-chemical analysis of the membranes is provided in the study. The inclusion of data on the mechanical properties (e.g. strenght, elongation and Young's modulus), and thermogravimetric (TG/DTG), calorimetric (DSC) and spectroscopic profile would be very helpful for the proper characterization of the membranes.

 3) The results of the current study are well substantiated, justified and discussed. . However, some parts of the discussion are verbose and repetitive. For example, the first paragraph brings information already mentioned in the introduction, and could be completely deleted.

 4) Some brief report on the limitations of the study should be added to the manuscript.

Author Response

Point-by-point reply to Reviewer #2

Dear editor

 The study addressed at the potential use of a ECM fibrilar-derived 2D membrane extracted from C. reniformis (marine sponges) for wound healing applications. For this purpose, the authors assessed the biodegradability, permeability to water, proteins, and bacteria, and cellular response in fibroblast cell lines of the membranes. The data obtained in this study show na interesting extraction yield of approximately 19%, and suggest that the membrane is almost impermeable to proteins and can work as a efficient barrier against liquid loss and bacteria infiltration, although it does not affect the growth rate of bacteria commonly found in human skin surfaces. The 2D-CM is also able to modulate ECM production-/remodelling-related genes expression, such as up-regulation fibronectin and down-regulation of TGF-β genes. The study is relevant and well designed, with promising results for the further development of wound healing in vivo studies and clinical trials.

 Here are my comments and suggestions.

  • Some relevant information needs to be added to the abstract, such as results related to potential ECM production-/remodelling-related genes modulation.

R: We thank the reviewer for pointing out this aspect. As suggested, we added the missing information to the abstract.

  • In the introduction section, the authors have mentioned that there is still gap of knowledge about the molecular characterization and the mechanisms underlying the biosynthesis of the biopolymers that have been reported in wound healing studies. However, no physical-chemical analysis of the membranes is provided in the study. The inclusion of data on the mechanical properties (g. strength, elongation and Young's modulus), and thermogravimetric (TG/DTG), calorimetric (DSC) and spectroscopic profile would be very helpful for the proper characterization of the membranes.

R: For Chondrosia reniformis, a complete characterization of the gene sequences coding for its collagen is still missing. However, physical-chemical analyses of the xerogel that can be produced starting from this biomaterial have been already performed by the authors in previous works that are cited in the manuscripts:

  • Pozzolini, M.; Scarfì, S.; Gallus, L.; Castellano, M.; Vicini, S.; Cortese, K.; Gagliani, M.C.; Bertolino, M.; Costa, G.; Giovine, M. Production, Characterization and Biocompatibility Evaluation of Collagen Membranes Derived from Marine Sponge Chondrosia Reniformis Nardo, 1847. Marine Drugs 2018, 16, 111, doi:10.3390/md16040111.
  • Tassara, E.; Orel, B.; Ilan, M.; Cavallo, D.; Dodero, A.; Castellano, M.; Vicini, S.; Giovine, M.; Pozzolini, M. Seasonal Molecular Difference in Fibrillar Collagen Extracts Derived from the Marine Sponge Chondrosia Reniformis (Nardo, 1847) and Their Impact on Its Derived Biomaterials. Marine Drugs 2023, 21, 210, doi:10.3390/md21040210.
  • The results of the current study are well substantiated, justified and discussed. However, some parts of the discussion are verbose and repetitive. For example, the first paragraph brings information already mentioned in the introduction and could be completely deleted.

R: In response to the reviewer's suggestion, we have condensed the initial portion of the discussion, retaining only a few phrases that we deem relevant for summarizing the content and introducing the discussion.

  • Some brief report on the limitations of the study should be added to the manuscript.

R: The main limitations of the study have been reported in the conclusions paragraph, as suggested.

Reviewer 3 Report

The submitted work has significant shortcomings in methodological terms and cannot be accepted for publication in its current form. 

Main remarks:

1) The authors stand all the evidence base of their work based on the fact that they work with "collagen". However, the method described by them allows to obtain at best collagen hydrolysate. At a minimum, they need to do a biochemical analysis of the sponge extract they obtained, at least SDS-PAGE electrophoresis, and ideally a complete analysis of the amino acid composition. Conduct functional tests to prove that the resulting protein is collagen - the ability to form fibrils and hydrogels.

2) The results on membrane permeability to water and proteins are highly doubtful. This is especially important in the context of a possible application, indicated by the authors, for regenerative medicine. Materials with such properties are usually almost incapable of permeation by cells and blood vessels.

3) The experiment with cells also raises questions. It should be supplemented with at least a live-dead test at different times and a test for evaluation of proliferative activity and cell morphology.

Author Response

Point-by-point reply to Reviewer #3

The submitted work has significant shortcomings in methodological terms and cannot be accepted for publication in its current form. 

Main remarks:

  1. The authors stand all the evidence base of their work based on the fact that they work with "collagen". However, the method described by them allows to obtain at best collagen hydrolysate. At a minimum, they need to do a biochemical analysis of the sponge extract they obtained, at least SDS-PAGE electrophoresis, and ideally a complete analysis of the amino acid composition. Conduct functional tests to prove that the resulting protein is collagen - the ability to form fibrils and hydrogels.

R: Due to the peculiar properties of C. reniformis’ collagen, the extraction method that we used allowed us to obtain intact collagen fibers and not a collagen hydrolysate: this approach has been well-documented in a previously published paper, which serves as a foundational reference for the current work (that aims to provide a more comprehensive analysis of the topic). The cited paper has been referenced multiple times throughout the manuscript:

Pozzolini, M.; Scarfì, S.; Gallus, L.; Castellano, M.; Vicini, S.; Cortese, K.; Gagliani, M.C.; Bertolino, M.; Costa, G.; Giovine, M. Production, Characterization and Biocompatibility Evaluation of Collagen Membranes Derived from Marine Sponge Chondrosia Reniformis Nardo, 1847. Marine Drugs 2018, 16, 111, doi:10.3390/md16040111.

In this paper, the authors produced both SEM and TEM images of the C. reniformis’ extract, proving that it is in fact made up of intact collagen fibers, and subjected this biomaterial to several biochemical and chemical-physical tests.

Additionally, the physical-chemical features of C. reniformis’ collagen extract have been already described years ago also in the works of Garrone et al. ad Imhoff et al., which also contain the first aminoacidic profile of this biopolymer:

Garrone, R.; Huc, A.; Junqua, S. Fine Structure and Physicochemical Studies on the Collagen of the Marine Sponge Chondrosia Reniformis Nardo. Journal of Ultrastructure Research 1975, 52, 261–275, doi:10.1016/S0022-5320(75)80117-1.

Imhoff, J.M.; Garrone, R. Solubilization and Characterization of Chondrosia Reniformis Sponge Collagen. Connective Tissue Research 1983, 11, 193–197, doi:10.3109/03008208309004855

A more recent aminoacidic profile of the extract is reported in the following paper (also cited in the manuscript): 

Tassara, E.; Orel, B.; Ilan, M.; Cavallo, D.; Dodero, A.; Castellano, M.; Vicini, S.; Giovine, M.; Pozzolini, M. Seasonal Molecular Difference in Fibrillar Collagen Extracts Derived from the Marine Sponge Chondrosia Reniformis (Nardo, 1847) and Their Impact on Its Derived Biomaterials. Marine Drugs 2023, 21, 210, doi:10.3390/md21040210.

This paper also contains biochemical evaluations of the material, as well as SEM images and DMA/DMTA and DSC analyses of the biomaterial.

  1. The results on membrane permeability to water and proteins are highly doubtful. This is especially important in the context of a possible application, indicated by the authors, for regenerative medicine. Materials with such properties are usually almost incapable of permeation by cells and blood vessels.

R: The authors acknowledge that the collagen suspension used in this study, at the concentration chosen based on their previous published papers, resulted in a xerogel that exhibited high impermeability to water and proteins. However, as mentioned in line 288, the porosity and permeability of the xerogel can be modulated by diluting the starting material. In this study, the aim was to provide a more comprehensive characterization of the previously presented 2D-CMs, utilizing the same initial concentration as in the previous paper. Therefore, a different dilution was not tested, considering that the primary purpose of this biomaterial was wound isolation and exudate containment rather than cell infiltration.

  1. The experiment with cells also raises questions. It should be supplemented with at least a live-dead test at different times and a test for evaluation of proliferative activity and cell morphology.

R: Biocompatibility and proliferation tests on this material have been already performed in these papers, previously published by the authors:

Pozzolini, M.; Scarfì, S.; Gallus, L.; Castellano, M.; Vicini, S.; Cortese, K.; Gagliani, M.C.; Bertolino, M.; Costa, G.; Giovine, M. Production, Characterization and Biocompatibility Evaluation of Collagen Membranes Derived from Marine Sponge Chondrosia Reniformis Nardo, 1847. Marine Drugs 2018, 16, 111, doi:10.3390/md16040111.

Tassara, E.; Orel, B.; Ilan, M.; Cavallo, D.; Dodero, A.; Castellano, M.; Vicini, S.; Giovine, M.; Pozzolini, M. Seasonal Molecular Difference in Fibrillar Collagen Extracts Derived from the Marine Sponge Chondrosia Reniformis (Nardo, 1847) and Their Impact on Its Derived Biomaterials. Marine Drugs 2023, 21, 210, doi:10.3390/md21040210.

 Round 2

Reviewer 1 Report

I have no more questions.

Reviewer 2 Report

Dear editor

All suggestions and requests made to the authors of the manuscript were satisfactorily answered. Thus, the manuscript shows overall improvement , particularly in scientific soundness. For this reason, I recommend that the manuscript be accepted for publication.

Reviewer 3 Report

I thank the authors for providing the articles, it allowed me to clarify some points. I can correct some of my comments.

1) The material the authors are working with is a crude biomaterial containing collagen. The amount of proteins other than collagen in this material has not been studied. This is especially important in the context of the potential use of this material for skin regeneration. SDS-PAGE electrophoresis of the resulting extract could answer this question.

2) So far, the comment about the ability of this collagen-containing material to form hydrogels at neutral pH values and temperature of +37C has remained unanswered. Rheological tests can answer this question (G’ and G” value as function of time at +37 °C).

3) About permeability. It is not correct to use Xerogel in this case to compare permeability for water and proteins. It should be compared with collagen membrane obtained from highly purified preparation of mammalian collagen, for example.

4) Given that the material obtained is almost impermeable to proteins and water test for bacterial penetration is not representative. Any other materials that are impermeable to water - polyethylene membranes, for example - have a similar property. However, given that the authors are positioning their development for skin regeneration, the designated properties of impermeability to water raise strong concerns. In the process of natural healing, exudative fluid is formed and if its outflow from the wound is blocked, severe aseptic inflammation will occur.

5)By a bioresorption test using Clostridium histolyticum.

This enzyme usually totally degrades collagen biomaterial prepared from highly purified collagen in a few hours. The fact that in this work the material is not degrade in more than a week under the action of this enzyme, raises even more questions about the composition of the material

Author Response

Answers for reviewer 3

While thanking the reviewer for the further remarks, to avoid further doubts, the authors renew our invitation to read the references describing the fibrillar collagen extract from C. reniformis:

Imhoff, J.M.; Garrone, R. Solubilization and Characterization of Chondrosia Reniformis Sponge Collagen. Connective Tissue Research 1983, 11, 193–197, doi:10.3109/03008208309004855

Berillis, P. Marine Collagen: Extraction and Applications. (2015)

Pozzolini, M.; Scarfì, S.; Gallus, L.; Castellano, M.; Vicini, S.; Cortese, K.; Gagliani, M.C.; Bertolino, M.; Costa, G.; Giovine, M. Production, Characterization and Biocompatibility Evaluation of Collagen Membranes Derived from Marine Sponge Chondrosia Reniformis Nardo, 1847. Marine Drugs 2018, 16, 111, doi:10.3390/md16040111

Tassara, E.; Orel, B.; Ilan, M.; Cavallo, D.; Dodero, A.; Castellano, M.; Vicini, S.; Giovine, M.; Pozzolini, M. Seasonal Molecular Difference in Fibrillar Collagen Extracts Derived from the Marine Sponge Chondrosia Reniformis (Nardo, 1847) and Their Impact on Its Derived Biomaterials. Marine Drugs 2023, 21, 210, doi:10.3390/md21040210.

Moreover, before responding to all of the reviewer’s doubts and questions, we want to clarify the following aspect: the fibrillar collagen isolated from C. reniformis and used in this work possesses distinct attributes that set it apart from collagens found in mammals or fish. In our study, we do not work with highly purified tropocollagen. In other words, we do not have tropocollagen in an acidic solution and we do not proceed to create fibrils in vitro by transitioning to a neutral pH, as is commonly done with most collagens. Instead, we directly extract an intact collagen fibers suspension from the sponge extracellular matrix (ECM).

Our goal was to address the following question: considering that C. reniformis has a unique collagen (already described, see references), and considering that we have created 2D membranes with specific characteristics that have been thoroughly analyzed chemically and physically (Pozzolini et al., 2018), could these membranes, which possess the same features as those previously examined, serve as a viable solution for skin-like substitute (specific application reference)?

Stated these points, here our point-by-point answers are listed below:

  • The material the authors are working with is a crude biomaterial containing collagen. The amount of proteins other than collagen in this material has not been studied. This is especially important in the context of the potential use of this material for skin regeneration. SDS-PAGE electrophoresis of the resulting extract could answer this question.

Throughout the extraction process, certain ECM proteins and proteoglycans may persist within the biomaterial, potentially being crosslinked to the collagen fibers and therefore not easy to be separated. However, without conducting a proteomic analysis, we cannot determine the precise composition of these proteinaceous components. This particular aspect will be addressed in a separate study. However, an SDS-PAGE analysis of this biomaterial has already been presented in:

Pozzolini, M.; Millo, E.; Oliveri, C.; Mirata, S.; Salis, A.; Damonte, G.; Arkel, M.; Scarfì, S. Elicited ROS Scavenging Activity, Photoprotective, and Wound-Healing Properties of Collagen-Derived Peptides from the Marine Sponge Chondrosia reniformis. Mar. Drugs 2018, 16, 465. https://doi.org/10.3390/md16120465

The authors also tested biocompatibility in vitro in this work:

Pozzolini, M.; Scarfì, S.; Gallus, L.; Castellano, M.; Vicini, S.; Cortese, K.; Gagliani, M.C.; Bertolino, M.; Costa, G.; Giovine, M. Production, Characterization and Biocompatibility Evaluation of Collagen Membranes Derived from Marine Sponge Chondrosia reniformis Nardo, 1847. Mar. Drugs 2018, 16, 111. https://doi.org/10.3390/md16040111

 By morphology observation, no inflammation process in cells has been reported in this previous research and we proceeded with further analyses in this work where it was confirmed by no variation in IL-1 gene expression considering this information enough for an in vitro evaluation.

  • So far, the comment about the ability of this collagen-containing material to form hydrogels at neutral pH values and temperature of +37C has remained unanswered. Rheological tests can answer this question (G’ and G” value as function of time at +37 °C).

At neutral pH, this biomaterial is a highly insoluble hydrogel that has already been subjected to rheological tests in:

Pozzolini, M.; Scarfì, S.; Gallus, L.; Castellano, M.; Vicini, S.; Cortese, K.; Gagliani, M.C.; Bertolino, M.; Costa, G.; Giovine, M. Production, Characterization and Biocompatibility Evaluation of Collagen Membranes Derived from Marine Sponge Chondrosia reniformis Nardo, 1847. Mar. Drugs 2018, 16, 111. https://doi.org/10.3390/md16040111

Tassara, E.; Orel, B.; Ilan, M.; Cavallo, D.; Dodero, A.; Castellano, M.; Vicini, S.; Giovine, M.; Pozzolini, M. Seasonal Molecular Difference in Fibrillar Collagen Extracts Derived from the Marine Sponge Chondrosia Reniformis (Nardo, 1847) and Their Impact on Its Derived Biomaterials. Marine Drugs 2023, 21, 210, https://doi:10.3390/md21040210

In these studies, rheological tests were conducted at 20 °C. However, the ability of this collagen-based biomaterial to form hydrogels and its related rheological properties at 37 °C has been observed in the same material:

Fassini, D.; Duarte, A.R.C.; Reis, R.L.; Silva, T.H. Bioinspiring Chondrosia Reniformis (Nardo, 1847) Collagen-Based Hydrogel: A New Extraction Method to Obtain a Sticky and Self-Healing Collagenous Material. Marine Drugs 2017, 15, 380, doi:10.3390/md15120380.

  • About permeability. It is not correct to use Xerogel in this case to compare permeability for water and proteins. It should be compared with collagen membrane obtained from highly purified preparation of mammalian collagen, for example.

The authors did not compare the membranes with Xerogel. In the previous answer, the authors spoke about a xerogel indicating the membranes themselves, using this term as a generic word for indicating a hydrogel that was let ambient drying, as stated in:

Heinemann, S.; Coradin, T.; Desimone, M.F. Bio-Inspired Silica–Collagen Materials: Applications and Perspectives in the Medical Field. Biomater. Sci. 2013, 1, 688–702, doi:10.1039/C3BM00014A.

We acknowledge that the word “xerogel” may not have been used properly in our previous response.

  • Given that the material obtained is almost impermeable to proteins and water test for bacterial penetration is not representative. Any other materials that are impermeable to water - polyethylene membranes, for example - have a similar property. However, given that the authors are positioning their development for skin regeneration, the designated properties of impermeability to water raise strong concerns. In the process of natural healing, exudative fluid is formed and if its outflow from the wound is blocked, severe aseptic inflammation will occur.

The objective of this study was to assess the potential of a previously described biomaterial as device for skin injury. To achieve this, we utilized membranes with identical characteristics to those employed in previous studies (25x28 mm membranes produced through the drying of a 3 mg/mL collagen suspension). The experimental approach was also defined drawing inspiration from Ferrario et al., 2021 who successfully created and tested membranes using a sea urchin collagen suspension of similar concentration.

During our investigation, we discovered that the chosen concentration (3 mg/mL) resulted in membranes with a high frame density, impeding the passage of liquids. This observation, as highlighted by the reviewer, prompts the authors to focus on addressing this particular aspect to enhance the suitability of the biomaterial in further studies regarding its application. However, we proceeded to clarify this aspect better in the discussion and in the conclusions.

  • By a bioresorption test using Clostridium histolyticum.

This enzyme usually totally degrades collagen biomaterial prepared from highly purified collagen in a few hours. The fact that in this work the material is not degrade in more than a week under the action of this enzyme, raises even more questions about the composition of the material.

As mentioned before, in the process of extraction, together with the sponge intact collagen fibers we co-extract other ECM elements, such as glycosaminoglycans intimately associated with the collagen fibers. The presence of glycosaminoglycans in association with fibers could be one of the main causes of the resistance of this biomaterial toward collagenase degradation, as is known that GAGs can confer higher resistance versus this enzyme, see for instance:

Zhong; Teo, W.E.; Zhu, X.; Beuerman, R.; Ramakrishna, S.; Yung, L.Y.L. Formation of Collagen−Glycosaminoglycan Blended Nanofibrous Scaffolds and Their Biological Properties. Biomacromolecules 2005, 6, 2998–3004, doi:10.1021/bm050318p.